# Exploring Under Constraints with Model-Based Actor-Critic and Safety Filters

**Ahmed Agha**
Volkswagen Group
Germany
`ahmed.agha@volkswagen.de`

**Baris Kayalibay**
Volkswagen Group
Germany
`baris.kayalibay@volkswagen.de`

**Atanas Mirchev**
Volkswagen Group
Germany
`atanas.mirchev@volkswagen.de`

**Patrick van der Smagt**
Volkswagen Group
Germany

**Justin Bayer**
Volkswagen Group
Germany
`justin.bayer@volkswagen.de`

**Abstract:** Applying reinforcement learning (RL) to learn effective policies on physical robots without supervision remains challenging when it comes to tasks where safe exploration is critical. Constrained model-based RL (CMBRL) presents a promising approach to this problem. These methods are designed to learn constraint-adhering policies through constrained policy optimization approaches. Yet, such policies often fail to meet stringent safety requirements during learning and exploration. Our solution "CASE" aims to reduce the instances where constraints are breached during the learning phase. Specifically, CASE integrates techniques for optimizing constrained policies and employs planning-based safety filters as backup policies, effectively lowering constraint violations during learning and making it a more reliable option than other recent constrained model-based policy optimization methods.

**Keywords:** Model-based RL, Safe RL, Safety Filter, Exploration

## 1 Introduction

Many real-world robotic systems can be effectively modeled as constrained Markov decision processes (CMDPs) [1], particularly in contexts where safety is paramount, and robots must adhere to specific conditions while learning to accomplish tasks. In addition, CMDPs offer a structured framework for injecting inductive biases into the policy optimization process, thereby reducing the number of interactions required to learn effective policies. Thus, developing deep RL algorithms for solving CMDPs has the promise of unlocking RL's potential across various real-world robotic applications where safety is an issue. However, applying RL to systems that must operate under constraints all the time, even as they explore and learn, remains an open challenge.

Model-free constrained policy optimization methods have primarily adapted actor-critic techniques to the constrained setting, employing Lagrangian relaxation, such as the augmented Lagrangian method, to train constrained policies. While these algorithms are appealing for their relative simplicity, they face significant challenges, particularly in terms of sensitivity to Lagrangian multipliers [2]. Moreover, model-free RL approaches for CMDPs often suffer from high sample complexity, making them ill-suited for constrained environments where a high number of interactions during

8th Conference on Robot Learning (CoRL 2024), Munich, Germany.

learning constrained policies leads to a high number of constraint violations as well as increased wear and tear, reducing their suitability for learning without supervision on physical robots.

Various authors have proposed applying model-based RL methods to constrained settings to tackle the challenge posed by the high sample complexity of model-free approaches. These methods fall into two main categories. The first category includes methods that improve learning efficiency and reduce constraint violations during exploration by using differentiable models [3, 4]. These models act as simulators for both learning the policy and the value function. However, despite their advantages, they often face difficulties with the instability that comes with constrained policy optimization. The second category involves methods that incorporate planning [5, 6, 7, 8] possibly alongside the primary, unconstrained policies. This approach helps in preventing constraint violations throughout the learning phase. We find that combining the ability of learned models to act as simulators for constrained policy optimization with the possibility of using learned models in look-ahead control schemes remains a largely unexplored direction. Furthermore, we find that most CMBRL methods do not consider epistemic uncertainty, which is essential in cases where the policy is exploring online and needs to reason about the effect of epistemic uncertainty on its ability to satisfy constraints.

**Our Contributions**  In this work, we combine constrained model-based policy optimization with a planning-based safety filter that acts as a backup policy to minimize constraint violations during exploration. In addition, we introduce modifications to the constrained model-based policy optimization training to ensure stable training. We also modify the safety filter's objective to consider the behavior of the constrained base policy during planning. We evaluate our method on constrained tasks from the Omnisafe [9], RWRL [10], and the safe-control-gym [11] benchmarks and show that our combination of constrained policy optimization and planning can lead to significantly reducing constraint violations during training in comparison with other model-free and model-based methods.

## 2 Preliminaries

### 2.1 Problem Setting

We consider a CMDP, which is defined by the tuple $(\mathcal{S}, \mathcal{A}, p, \rho_0, r, \gamma, c, \gamma_{\text{safe}})$. $\mathcal{S}$ and $\mathcal{A}$ are the state and action spaces respectively. The discount factor and the safety discount factor are represented by $\gamma$ and $\gamma_{\text{safe}}$. The dynamics of the system are represented by $p(s_{t+1} \mid s_t, a_t)$, and the initial state distribution is represented by $\rho_0$. The reward function is represented by $r(s_t, a_t)$, and the constraint cost function is represented by $c(s_t)$. In this work, we consider constraint costs represented with the indicator function $\mathcal{I}(s_t)$ where the unsafe state is represented as $\mathcal{S}_{\text{unsafe}} = \{s_t \mid \mathcal{I}(s_t) = 1\}$ and the safe states are represented as $\mathcal{S}_{\text{safe}} = \{s_t \mid \mathcal{I}(s_t) = 0\}$.

The challenge in CMDPs is maximizing the performance of the agent while satisfying constraints

$$J(\pi) = \mathop{\mathbb{E}}_{a_t \sim \pi, s_0 \sim \rho_0} \left[ \sum_{t=0}^{\infty} \gamma^t r(s_t, a_t) \right] \quad \text{such that} \quad J^c(\pi) = \mathop{\mathbb{E}}_{a_t \sim \pi, s_0 \sim \rho_0} \left[ \sum_{t=0}^{\infty} \gamma_{\text{safe}}^t \mathcal{I}(s_t) \right] \leq l, \quad (1)$$

where $l$ is a problem-specific threshold.

### 2.2 Constrained Model-based Reinforcement Learning

In constrained model-based RL, we use transition tuples in the form of $\{s_t, a_t, r_t, c_t, s_{t+1}\}$ to learn a transition model $p$. The transition model can then be used in online planning as done in [12, 13], or by amortizing decision-making by offline training of a parametric policy using an actor-critic approach as in [14, 15].

Recent methods [3, 4] have shown the effectiveness of MBRL in solving CMDPs as formulated in Eq. (1). Both methods adapt previous model-based RL methods to the constrained setting; LAMBDA [3] is based on Dreamer [14], where the RSSM model from PLANET [13] is used as a differentiable simulator for on-policy actor-critic training to learn a policy $\pi_\theta$ and a critic $v^\pi$;

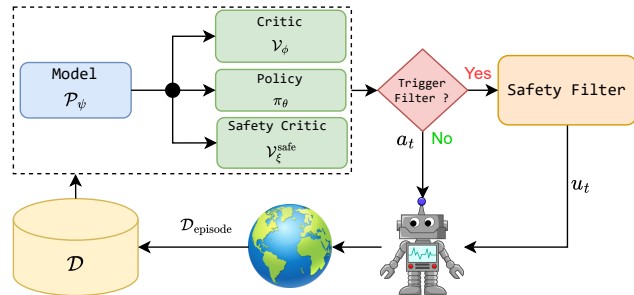

Figure 1: Overview of CASE. A dynamics model is used to train a constrained policy, a critic, and a constraint critic. The task policy is used alongside a safety filter to ensure low constraint violation rates during exploration.

safeSLAC [4] is based on SLAC [16], where the model is used to fill a replay buffer used in an off-policy algorithm. Both methods extend their base MBRL algorithm by adding a constraint critic $v^{\pi,\text{safe}}$ and using a Lagrangian relaxation to train a constrained policy by optimizing

$$\min_{\lambda \geq 0} \max_{\pi_\theta} \mathbb{E}_{s_0 \sim \rho_0}[J(s_0)] + \lambda \mathbb{E}_{s_0 \sim \rho_0}[J^c(s_0)]. \tag{2}$$

We follow a similar constrained model-based actor-critic (CMBAC) scheme to LAMBDA. However, we do not follow a similarly complex approach to the multiplier updates; rather, we add some minor modifications to the calculation of the value functions in constrained policy optimization to ensure the stability of the multiplier updates using gradient descent. Additionally, we avoid learning the policy $\pi_\theta$ using a pessimistic constraint cost, which leads to learning an overtly conservative policy [4].

## 3   Approach

In CASE, we combine CMBAC with a planning-based safety filter to lower constraint violation rates while exploring CMDPs. Our motivation is to leverage the advantages of look-ahead control and the computational efficiency of actor-critic methods to explore the CMDP while keeping constraint violations to a minimum. This combination of learning and planning has been studied in recent work [6, 17, 18], leveraging the ability of actor-critic methods to learn parametric policies and critics efficiently and compensating for the bias and limited expressiveness of an amortized parametric policy with look-ahead control using the model. We learn a probabilistic ensemble transition model $\mathcal{P}_\psi$ similar to that introduced in PETS [12], which we discuss further in appendix B. We use

---

**Algorithm 1** Pseudocode of CASE

Initialize parameters $\theta, \phi, \xi, \psi$
**for** N episodes **do**
  **for** t timesteps **do**
    sample $a_t$ from policy $\pi_\theta(s_t)$
    **if** $s_t \in \mathcal{S}_{\text{recovery}}$ according to (4) **then**
      Trigger filter and optimize filter objective
      in (5) resulting in plan $\{u_t, \ldots, u_{t+H}\}$
      **Apply** $u_t$
    **else**
      **Appy** $a_t$
    **end if**
  **end for**
  Add $\mathcal{D}_{\text{episode}}$ to $\mathcal{D}$
  update $\mathcal{P}_\psi, \pi_\theta, \mathcal{V}_\phi, \mathcal{V}_\xi^{\text{safe}}$
**end for**

---

the learned dynamics model in a model-based actor-critic algorithm as a learned simulator for constrained on-policy actor-critic training. Online exploration is performed primarily using the constrained parametric policy; in addition, we implement a pessimistic planning-based safety filter to avoid violating constraints. An overview of our methods can be seen in figure 1 and algorithm 1.

## 3.1 Constrained Model-based Actor-Critic

Our approach to policy optimization is most similar to [3, 4], where CMBAC methods were shown to result in data-efficient learning of constrained policies. For the policy optimization, we learn a parameterized policy $\pi_\theta$. In addition, we follow [19] and learn an ensemble value function $\mathcal{V}_\phi = \{v_{\phi_1}, \ldots, v_{\phi_B}\}$, where $v_{\phi_i}$ are individual ensemble members, similarly, we learn an ensemble safety critic $\mathcal{V}_\xi^{\text{safe}} = \{v_{\xi_1}^{\text{safe}}, \ldots, v_{\xi_B}^{\text{safe}}\}$.

**Learning Dynamics Model** We adapt the probabilistic ensembles from PETS [12] for solving CMDPs. We add predictions heads for rewards $p(r_t|s_t)$, constraint cost $p(\mathcal{I}_t|s_t)$, and termination flags $p(d_t|s_t)$ for environments with early termination conditions. We model the reward distribution $p(r_t|s_t)$ as a Gaussian distribution, while the binary constraint cost $\mathcal{I}_t$ and termination flag $d_t$ are modeled as Bernoulli distributions.

**Learning Critics** For learning of value function ensemble members, we use imagined rollouts $\tau_{i,s_t}$ using respective transition ensemble members $p_{\psi_i}$ and branching off real states $s_t$. This approach leads to the disagreement of critic ensemble members capturing the epistemic uncertainty in the transition function ensemble $\mathcal{P}_\psi$ similar to the approach followed in [20]. We use TD($\lambda$) to calculate the targets for the value function similar to Dreamer [14] and train each member in the value function ensemble on its own independent targets as done in [21]

$$\min_{\phi_i} \mathbb{E}_{s_{t'}\sim p_{\psi,i}, a_{t'}\sim\pi_\theta} \left[\sum_{t'=t}^{t+H_v} \frac{1}{2}\left\|v_{\phi,i}(s_{t'}) - R^i(s_{t'})\right\|^2\right] \text{, where}$$

$$R^i(s_t) = r_t + \gamma(1-d_t)\left((1-\lambda^{\text{reward}})v_{\phi,i}(s_t) + \lambda^{\text{reward}}R^i(s_{t+1})\right) , \ R^i_{t+H_v} = v_{\phi,i}(s_{t+H_v}).$$

Similarly we learn the safety critic ensemble $\mathcal{V}_\xi^{\text{safe}}$ using TD($\lambda$) targets We note that the termination flag $d_t$ is only included in calculating the task critic targets. We do not include it in the constraint critic's training, which would lead the constraint critic to underestimate the cost for states near termination states. We discuss this design choice further in the ablation studies in appendix A.

**Constrained policy optimization** For solving the constrained problem in equation (1), we resort to Lagrangian relaxation by including the constraint cost term in the policy objective weighted by the Lagrangian multiplier $\lambda$, thus turning the problem into an unconstrained problem as in equation (2), where we solve a min-max optimization over the policy $\pi_\theta$ and the Lagrangian multiplier $\lambda$

$$\min_{\lambda\geq 0}\max_{\pi_\theta} \frac{1}{1+\lambda}\mathbb{E}_{a_{t'}\sim\pi, s_t\sim\mathcal{S}_{\text{safe}}, s_{t'}\sim\mathcal{P}_\psi}\left[R(s_{t+k}) + \lambda C(s_{t+k}) \mid s_t\right], \tag{3}$$

where $R$ and $C$ are the means over the TD($\lambda$) returns from the different ensemble members $R^i$ and $C^i$ for the rewards and the constraint costs respectively. Designing stable update rules for Lagrangian multipliers is a challenging task in the model-based setting, as using biased model rollouts for updating the multipliers can lead to a rapid increase in their magnitudes, thus derailing training. Furthermore, model rollouts in MBRL normally use observations from the replay buffer as starting states, which can lead to the multipliers being updated to reflect the behavior of the policy used to collect the data rather than the optimized policy $\pi_\theta$. In [3], a complicated optimization scheme is used to decelerate the updates of the multipliers. In [4], the multipliers are updated solely using real online rollouts, presumably to avoid inaccuracies in the model from causing erroneous multiplier updates. We follow a more straightforward scheme and update the multipliers using stochastic gradient descent with no additional heuristics.

Our changes center around the calculation of the policy's objective and are highlighted in objective (3) in cyan. We only use constraint-satisfying states $\mathcal{S}_{\text{safe}}$ as initial states for our rollouts and only include the tail of the rollouts in the calculation of the terms in the objective, thus avoiding situations where the agent is already doomed but starts in a safe state. Thus giving the policy enough time to steer the system away from constraint-violating regions. Over time, these changes decelerate the increase in the multipliers and prevent them from exploding in value. In addition, we normalize the

whole objective by a factor of $1 + \lambda$, which helps keep the absolute value of the loss in the same scale as the multiplier $\lambda$ increases in value similar to [2].

## 3.2 Exploration with Safety Filter

Our aim is to enable safer online exploration. Thus, we do not use conservatism to prevent the policy from exploring online as in [22, 23, 3], which would lead the policy to learn an overtly conservative behavior. In contrast, we rely on a conservative safety filter MPC as a backup policy that intervenes to prevent constraint violation, guided by the critic.

**Trigger** To trigger the filter, we rely on the ensemble transition model and use it to roll out the learned policy $\pi_\theta$ for horizon $H^{\text{filter}}$ starting from the current state $s_t$, generating separate imagined trajectories $\{(s_{t'}^i, a_{t'}^i)\}_t^{t+H^{\text{filter}}}$ using each member $p_{\theta,i}$. We use the worst-case value of the ensemble safety critic $\mathcal{V}_{\text{max}}^{\text{safe}} = \max\limits_{s_{t'} \in \mathcal{P}_\psi} \mathcal{V}^{\text{safe}}(s_{t'})$ to define a pessimistic recovery set

$$\mathcal{S}_{\text{recovery}} = \{(s_t) \in \mathcal{S} : \mathcal{V}_{\text{max}}^{\text{safe}}(s_{t'}^i) \geq \epsilon_{\text{safe}}\}, \tag{4}$$

where $\mathcal{V}_{\text{max}}^{\text{safe}}$ is the maximum prediction of the ensemble critic across the look-ahead trajectories.

Our pessimistic formulation of the objective and the trigger of the safety filter consider the epistemic uncertainty inherent in the exploration task, where the filter is more likely to be triggered in situations with high epistemic uncertainty due to the effect of considering the worst case prediction of the ensemble safety critic.

We roll out separate trajectories starting from current state $s_t$ with each separate transition ensemble member $p_{\psi,i}$ and evaluate the states $s_{t'}^i$ in each trajectory with its respective safety critic $v_{\xi,i}$. The state $s_t$ is considered part of $\mathcal{S}_{\text{recovery}}$ in case the worst case prediction of $\mathcal{V}^{\text{safe}}$ of the imagined trajectories starting from $s_t$ exceeds the threshold $\epsilon_{\text{safe}}$.

**Optimization** The safety filter used in this paper solves the optimization problem

$$\min_{u_t \ldots u_{t+m}} \mathbb{E}_{\substack{s_{t+1..t+m} \sim \mathcal{P}_\psi(.|s_{t'}, u_{t'}) \\ s_{t+m+1..t+H} \sim \mathcal{P}_\psi(.|s_{t'}, \pi_\theta(s_{t'}))}} \left[ \sum_{t'=t}^{t+H_{\text{filter}}} \mathcal{V}_{\text{max}}^{\text{safe}}(s_{t'}) \right]. \tag{5}$$

This MPC-filtering approach is similar to the recovery RL method [7], which minimizes the safety critic along the look-ahead horizon, thus enabling a longer look-ahead at a reduced computational cost. In addition, we use pessimism in triggering the filtering and optimizing its objective. Thus taking epistemic uncertainty into consideration. Furthermore, our objective includes rollouts from the base policy $\pi_\theta$ in the objective, thus encouraging the MPC to drive the systems to regions of the state space where the base policy is predicted to keep the system safe. We optimize the objective in (5) using gradient descent where we optimize the whole term for the actions $\{u_t, \ldots, u_{t+m}\}$.

## 4 Results

### 4.1 Experimental Setup

We compare our method with model-free baselines on constrained locomotion tasks from the Omnisafe [9] benchmark and model-based baselines on the RWRL [10] and safe-control-gym [11] benchmarks. We run each method on four seeds and show the mean and the 95% confidence interval performance in figure 2. We explain our experimental setup and hyperparameters in more detail in appendix D.

### 4.2 Comparison to Model-free Baselines

For benchmarking against model-free baselines, we use the constrained velocity control tasks from the Omnisafe benchmark, where the goal is to solve locomotion tasks while maintaining the system below a maximum velocity. We choose different constrained RL baselines based on TD3 [24]

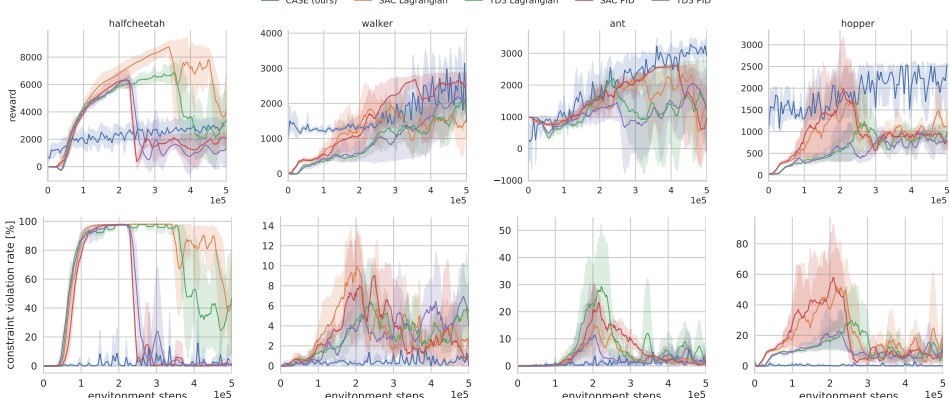

Figure 2: Performance of CASE compared to other constrained RL methods on constrained environments from the Omnisafe benchmark. We find that constraint violation rates during exploration are reduced significantly while maintaining competitive task performance in comparison with other constrained RL methods..

and SAC [25]. We first compare the performance of CASE with the model-free baselines SAC-Lagrangian and TD3-Lagrangian, which extend SAC and TD3 with a constrained policy optimization scheme using Lagrangian relaxation. In addition, we compare CASE to SAC-PID and TD3-PID, which use the Lagrange multipliers updating scheme presented in [2]. CASE generally reaches the same task performance as the model-free baselines while maintaining comparatively low constraint violation rates throughout training. However, we still see that even with the addition of a planner as a safety policy, we are still not able to completely avoid constraint violations.

The baselines eventually learn to solve the task while having low constraint violation rates. However, as constrained policy optimization tends to have instabilities during training, they have high rates of constraint violation during exploration. This is compounded by the implementation in Omnisafe, which only starts updating the Lagrange multipliers after 100 warm-up epochs, corresponding to 200k environment steps. This initial warm-up phase ensures the off-policy methods have enough variety in the replay buffer. In cases where the robot needs to learn in the wild, such exploration behavior might not be acceptable.

The results from our experiments put into question the suitability of the model-free method for safe reinforcement tasks, where the agent needs to learn online. Although model-free methods might have advantages in their asymptotic performance compared to model-based methods [12], they lack the ability of model-based methods to do look-ahead controlling and deciding online to avoid actions where the robot might be uncertain or that might be deemed to be possibly dangerous. Making model-based methods more suitable for learning on physical robots, especially when the robot needs to explore under certain restrictions.

### 4.3 Comparison to Model-based Baselines

We compare our method to two similar model-based approaches: LOOP [6] and the model-based version of Recovery RL [7].

LOOP integrates learned value functions with planning for decision-making. Unlike our method, LOOP uses look-ahead control to maximize rewards and minimize constraint violations at each step, does not account for epistemic uncertainty in planning, and assumes access to the ground-truth constraint cost function. Despite this, CASE outperforms or matches LOOP in constraint adherence without access to the real constraint cost function. Additionally, CASE demonstrates superior task performance, except in the cartpole environment, where LOOP has a higher constraint violation rate. As LOOP requires optimising a look-ahead control problem in every environment step, we find that

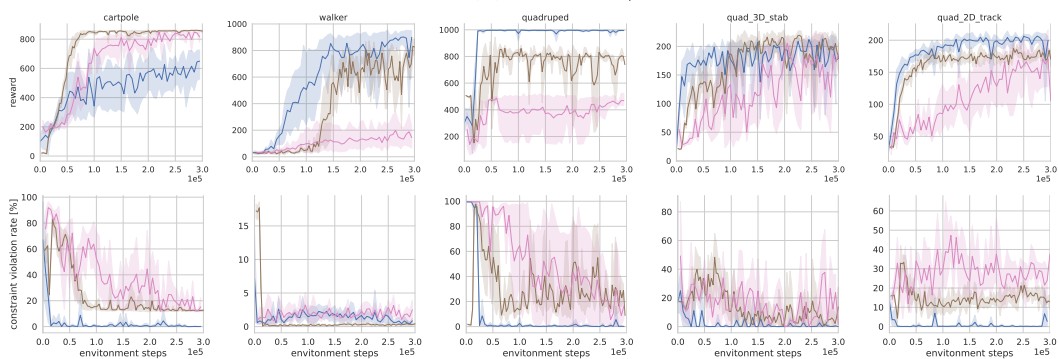

Figure 3: CASE outperforms both LOOP and Recovery RL in terms of constraint violations across constrained environments in the RWRL and Safe-Control-Gym benchmarks. Although CASE shows slightly higher violations in the walker environment compared to LOOP (still under 5%), it has the added challenge of predicting constraint costs using the learned model. In contrast, LOOP uses the environment's ground-truth constraint function to calculate costs, giving it an advantage in constraint adherence that CASE achieves without such prior knowledge.

it is much less compute efficient than CASE, we discuss the differences in frequency between CASE and similar methods in appendix C.

In contrast, Recovery RL uses a model-free approach to learn an unconstrained task-solving policy and a safety critic. The model-based version introduces a forward model for triggering interventions and to design a look-ahead recovery controller. However, Recovery RL suffers from conflicts between the task-solving policy and the recovery policy, leading to worse performance compared to CASE, as seen in figure 3.

## 5   Related Work

**Constrained model-based reinforcement learning**    Different RL methods have been introduced targeting CMDPs [26, 27, 28]. The most common variants are papers leveraging the augmented Lagrangian method for learning constrained policies. CPO [29] extended TRPO to CMDP, and building on that work [30] combined the Lagrangian method with PPO to learn constrained policy. Using an on-policy method rendered the method unsuitable for learning in the real world, where sample efficiency is essential. In [31], the authors implemented a similar approach but replaced the on-policy algorithm with an off-policy approach to reduce sample complexity. Saute RL [32] provides an alternative to using Lagrangian relaxation for solving CMDPs by using state space augmentation. Different papers proposed MBRL methods combining learned models with constrained policy optimization. In [3], the authors use a Bayesian approximation of an RSSM [13] as a model. The posterior of the transition is used to learn an optimistic objective regarding the rewards but pessimistic regarding the constraint cost. A similar approach was introduced in [4] where the authors extend SLAC [16] to the constrained setting. Concurrent with our work, safeDreamer proposes combining planning with CMBAC. However, SafeDreamer differs in not leveraging the planner as a backup policy, as is the case with CASE. Further, safeDreamer does not consider the epistemic uncertainty into decision-making, as proposed in our method.

**Safety filters**    Learning a policy to satisfy constraints and, at the same time, maximize the agent's utility can be a challenging task. As a result, multiple methods have attempted to circumvent this problem by introducing backup policies $\pi^{\text{safe}}(s_t)$ that minimize constraint costs in addition to a task-reward maximizing base policy $\pi(s_t)$. Model-free methods that use a backup mechanism [7, 33] generally use an off-policy critic to trigger the intervention mechanism. Critics tend to be sensitive to out-of-distribution data and, in general, hard to learn. The two-policy setup is also investigated

in [34], where the authors improve on the safety layers approach [35] by replacing the layers with a separate parametric policy that allows them to handle more complicated constraints.

The predictive safety filter paper [8] introduces an MPC wrapper to a base policy to prevent constraint violations during learning, relying on assumptions about the model and the system dynamics and the availability of a controller that keeps the system in a predetermined safe terminal state to have guarantees on constraint satisfaction. In [36], a control barrier function (CBF) is used in learning a safe policy, but their method is limited to control-affine systems and assumes prior knowledge of a simplified model and access to a forward invariant safe set. In general, our method differs from other safety filter methods based on CBFs and Hamilton-Jacobi reachability in that we do not make any assumptions on the underlying CMDP and do not assume prior knowledge of the nominal dynamics of the systems. This contrasts to CBF and HJ safety filters where it is common to make assumptions on the dynamics [37, 38, 39, 40, 36, 41], Lipschitz continuity [37, 38, 39, 40, 36] or existence of prior knowledge of prior safe sets [36]. SAILR [42] provides an advantage-based intervention mechanism and derives performance bounds under the assumption of having an MDP with an absorbing state. However, their model-based experiments seem to be based on engineered models.

**Online planning using offline learned functions**    Combining online planning with offline learned functions has proven effective in several works [18, 6, 17]. Online planning suffers less from bias compared to parameterized policies, and offline-learned functions can enhance the planner's performance [43]. The idea of planning using offline learned functions was shown to improve on parametric policies in [18], and performance bounds were derived that show the benefits of using a planner in combination with a value function. However, the methods and the bounds introduced all used an unbiased dynamics model. In [6], these insights were extended to learned models.

## 6    Conclusions and Limitations

We present a method that combines constrained model-based policy optimization with a pessimistic planning-based safety filter for exploration in CMDPs with the aim of facilitating the learning of effective policies entirely on physical robots. We find that leveraging the ability of model-based methods to function as a simulator for actor-critic methods and being used in look-ahead-control schemes, thus combining the advantages of model-based actor-critic with low-bias properties of planning, can lead to a significant reduction in constraint violation in comparison with model-free and model-based baselines. We evaluated our method on different constrained RL benchmarks, which show the versatility of our method in adapting to different task dimensionalities and its ability in robotic tasks such as drone trajectory tracking and stabilization.

Our method differs from previous safety filtering methods [8, 7], where the base policy $\pi_\theta$ is unaware of constraints and the agent's safety is left to the backup mechanism, often causing oscillatory behavior [44]. We assume no prior knowledge of the CMDP's dynamics and make no assumptions regarding its properties as done in previous safety filtering methods [36, 8, 39] and aim for a versatile method applicable to any CMDP, including those with contact dynamics discontinuities and high dimensional observations such as images. We believe combining our approach with other model-based actor-critic methods [45, 46] can enable efficient robot learning from high dimensional observations with minimal supervision.

**Limitations**    The main limitation of our method lies in the inability to guarantee constraint satisfaction. Such guarantees would involve making assumptions on the target system, such as the smoothness of the dynamics and the availability of a safe terminal set as in methods leveraging HJ reachability and CBF [8, 36, 40, 37, 38]. Such assumptions might not be satisfied in many target systems of interest. Although our method leads to less constraint violation than the baselines during exploration, online planning can be computationally expensive. Furthermore, our design for the learned transition model limits our method to fully observable systems, which we aim to rectify in future work by leveraging models that can learn the dynamics of partially observable systems.

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

# A    Ablation Studies

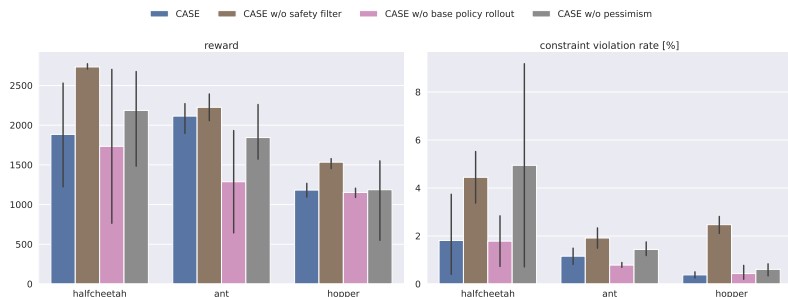

Figure 4: For ablating design choices in the safety filter. We evaluate the average rewards per episode in the last 10 evaluations (left) and the average percentage of steps with constraint violations in an episode during exploration (right). We first evaluate the role of the safety filter in CASE. Exploring with only $\pi_\theta$ leads to a significant increase in constraint violations. Next, we evaluate the effect of the base policy rollouts. We see a decrease in the average rewards reached after exploration, especially in the ant environment. Finally, to evaluate the effect of pessimism in the filter, we optimize the filter by using the mean over the ensemble predictions instead of considering the worst-case ensemble prediction. Without pessimism, the filter tends to have higher constraint violation rates.

**Effect of safety filter**    To assess the effect of the safety filter in our method, we explore CASE without the safety filter. In figure 4, we see that exploring solely using the constrained policy $\pi_\theta$ leads to higher constraint violation rates. This indicates the necessity of the filtering mechanism in our method. By looking at the maximum rewards reached after exploration in figure 4, we see that the increase in rewards after removing the safety filter is insignificant except in the halfcheetah environment, where the constraint violation rates more than double after removing the filter.

**Effect of base policy term in the filter objective**    The filter objective in (5) includes rolling out the base policy $\pi_\theta$ in the tail of the look-ahead trajectory of the controller. We ablate this term by replacing it at the tail of the look-ahead trajectory, thus maintaining a fixed total horizon length. This equates to the following objective

$$\min_{u_t\dots u_{t+H}} \mathbb{E}_{s_{t+1}..t+H \sim \mathcal{P}_\psi(.|s_{t'},u_{t'})} \left[ \sum_{t'=t}^{t+H} \mathcal{V}^{\text{safe}}_{\max}(s_{t'}) \right].$$

The purpose of the base-policy rollout is to inform the look-ahead controller of the base policy's behavior and thus steer the system towards regions of the state space where the base policy can perform without violating constraints. Removing the base policy rollouts leads to slightly worse task performance, especially in the ant environment. However, the difference in task performance is less significant than expected. In future work, exploring other possibilities for combining the intervention controller with a constrained base policy, such as penalizing the difference between the intervention mechanism's actions and those of the base policy as done in [8], might be interesting.

**Effect of pessimism in the safety filter**    In our safety filter design, we leverage a pessimistic loss to consider the effect of epistemic uncertainty during exploration. To study the effect of pessimism on the filter, we use the mean over the ensemble prediction constraint critic predictions $\mathcal{V}^{\text{safe}}_{\text{mean}}$, instead of using the maximum aggregation over the ensemble of the constraint critic predictions $\mathcal{V}^{\text{safe}}_{\max}$, introduced in section 3.2. We found that in our setup, removing the pessimism leads to higher constraint violation rates, especially in the more complex halfcheetah and ant environments.

**Effect of Including Termination flag in Constraint Critic's Targets**    The constraint critic target described in 3.1 does not include the termination flag $d_t$. The termination flag is usually needed in environments with termination conditions, which assumes that terminating states are absorbing

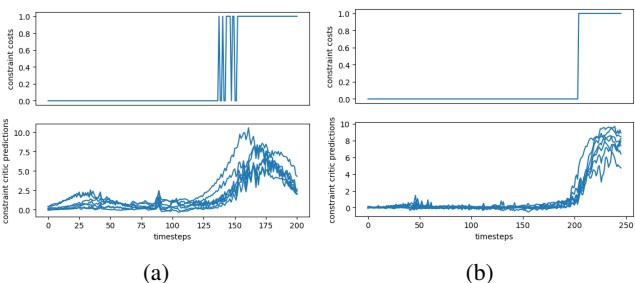

(a)                          (b)

Figure 5: We compare the predictions of the constraint critic when including the terminal flag in its target calculation in 5a and excluding the terminal flag in 5b for two different episodes with early termination. The top row shows the ground truth constraint costs, and the bottom row shows the constraint critic predictions for each state in the episodes. In 5a, we see the effect of adding the termination flags in the targets, where the critic predicts a lower constraint cost near the terminal state. Excluding the terminal state flag leads the constraint critic in 5b to avoid this effect.

|  | quad2D stab | quad2D track |
|---|---|---|
| LOOP | $12.01 \pm 0.86$ | $11.68 \pm 0.75$ |
| LOOP Slow | $0.94 \pm 0.01$ | $0.55 \pm 0.01$ |

Table 1: Comparing FPS for LOOP and LOOP Slow. We see that reducing the controller budget massively increases the FPS while having no effect on the performance, as seen in figure 7.

states. Including the termination flag in the task critic $\mathcal{V}$ leads the critic to predict low values for such states, incentivizing the policy to avoid termination. This effect is not desired in the constraint critic $\mathcal{V}^{\text{safe}}$ as it would lead the constraint critic to assign erroneous cost-to-gos to states near the termination states, which would have an adverse effect on the safety filter and the constrained policy optimization. We compare the predictions of two constraint critics in figure 5. The first in figure 5a is trained using targets including the termination flag $d_t$

$$C^i(s_t) = \mathcal{I}_t + \gamma^{\text{safe}}(1 - d_t)\left((1 - \lambda^{\text{safe}})v^{\text{safe}}_{\xi,i}(s_t) + \lambda^{\text{safe}}C^i(s_{t+1})\right) \quad \text{where,}$$
$$C^i(s_{t+H_v}) = v^{\text{safe}}_{\xi,i}(s_{t+H_v}).$$

The second in figure 5b is trained with the targets described in (5). We find that including $d_t$ in TD$(\lambda)$ returns has a negative effect on the constraint critic predictions where the critic assigns much lower values to states near the termination state, despite constraint violations.

## B   Deep Ensemble Transition Models

Due to our focus on exploration, we need a model that can provide well-calibrated epistemic uncertainties. Deep ensemble models present a straightforward approach to provide representations of the epistemic uncertainty due to their ability to provide good approximations of the Bayesian posterior predictive distribution of the neural network [47]. Thus, we rely on an ensemble of dynamic models $\mathcal{P}_\psi = \{p_{\psi,1}, \ldots, p_{\psi,\text{B}}\}$, where each ensemble member $p_{\psi,i}$ is a neural network that predicts the transition as a Gaussian distribution with a diagonal covariance $p_{\psi,i}(s_{t+1} \mid s_t, a_t) = \mathcal{N}(s_{t+1} \mid \mu_{\psi,i}(s_t, a_t), \Sigma_{\psi,i})$. We optimize each ensemble member by minimizing the negative log-likelihood

$$\mathcal{L}_{p_{\psi,i}} = -\sum_{t=1}^{T} \log p_{\psi,i}(s_{t+1} \mid s_t, a_t).$$

The use of ensemble-based transition models has already shown very good performance in model-based RL papers such as [48] and [12]. In deep ensemble transition models, each ensemble member $p_{\psi,i}$ captures the aleatoric uncertainty of the ground-truth MDP. In contrast, the disagreement between the ensemble members captures the epistemic uncertainty on the learned transition function.

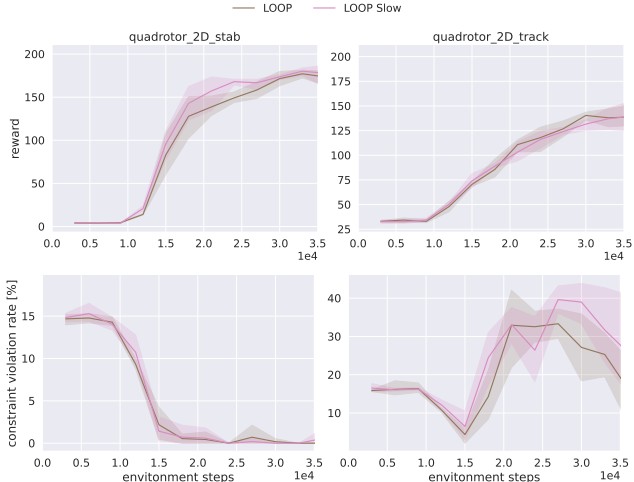

Figure 7: To reduce the runtime of LOOP seen in table 2, we tune the budget of the sampling-based look-ahead control used in LOOP. LOOP Slow is LOOP with the original hyperparameters, LOOP is the variant with the updated HPs. We see that both HP sets have similar results, and we see in table 1 the massive increase in FPS with the new HPs.

## C  Runtime

Although using a look-ahead planner as a safety filter helps reduce the constraint violation rates of the reactive parametric base policy, this comes at the cost of computational efficiency. The safety filter involves an online optimizer that solves the optimization problem in objective (5). We compare the frequency of the filter across the different environments in figure 6, and we find that the filter performs around 50 Hz in all environments. Looking at other methods combining actor-critic methods and planning, we find that TD-MPC [17], which does not consider the constrained setting, performs similarly to our method in runtime with about 50 Hz for the default setting. LOOP [6], which also explores CMDPs among other settings, reports a lower frequency 14.3 Hz. Generally, the low frequency of planning-based methods represents the biggest disadvantage compared to methods leveraging only parametric policies for decision-making.

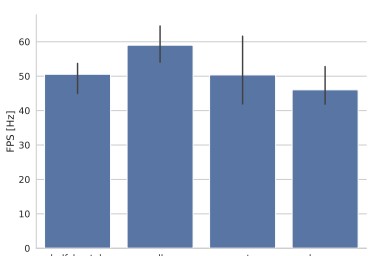

Figure 6: Average runtime for CASE. We find that despite the online computations, we maintain a reasonable FPS that makes CASE feasible to use in physical systems.

## D  Experimental Setup Details

### D.1  Benchmarks

We use the constrained velocity tasks from safety gymnasium [49] in our experiments. Specifically, we use the half cheetah, hopper, walker, and ant tasks. These tasks are attractive as they pose conflicting objectives, where the task reward incentivizes the agent to move with high velocity with the correct pose, and the constraint cost punishes the agent for moving above a velocity limit.

### D.2  Tuning LOOP for Safe Control Gym Tasks

In training LOOP [6] for the RWRL benchmark tasks, we used the hyperparameters (HPs) without tuning (we also do not tune our method). We noticed that the runtime for LOOP is extremely long and much worse than the latency mentioned in the paper when evaluated in high dimensional

|  | cartpole | walker | quadruped |
|---|---|---|---|
| CASE (Ours) | $70.5 \pm 3.3$ | $69.03 \pm 1.83$ | $79.7 \pm 7.8$ |
| LOOP | $7.9 \pm 0.13$ | $1.12 \pm 0.03$ | $0.88 \pm 0.016$ |

Table 2: FPS for CASE in comparison with LOOP [6]. We find that despite the online computations, we maintain a reasonable FPS that makes CASE feasible to use in physical systems.

environments, as seen in table 2. We found that this is mostly due to the loops in the sampling-based look-ahead controller in LOOP. To avoid the long LOOP training times in our experiments in safe-control-gym, we tuned the look-ahead controller aiming to decrease LOOP's latency while maintaining the same performance. We compare the performance of the new LOOP HPs with the original LOOP HPs on two of the safe-control-gym tasks in figure 7, and we compare their runtimes in table 1.

### D.3 Hyperparameters

Our method involves hyperparameters for the model training, the CMBAC, and the safety filtering. We list our hyperparameters in table 3.

|  | HalfCheetah | Hopper | Ant | Walker |
|---|---|---|---|---|
| **Model** $\mathcal{P}_\psi$ | | | | |
| number of bootstraps $B$ | | 7 | | |
| learning rate | | $1e^{-3}$ | | |
| activation | | softsign | | |
| number of hidden layers | | 4 | | |
| number of hidden units | | 200 | | |
| **Critic** $\mathcal{V}_\phi$ and **safety critic** $\mathcal{V}_\xi^{\text{safe}}$ | | | | |
| Horizon | | 12 | | |
| Activation | | softsign | | |
| TD $\lambda^{\text{reward}}$ | | 0.9 | | |
| discount $\gamma$ | | 0.99 | | |
| safety critic discount $\gamma^{safe}$ | | 0.9 | | |
| safety TD $\lambda^{safe}$ | | 0.75 | | |
| number of hidden layers | | 2 | | |
| number of hidden units | | 256 | | |
| **Policy** $\pi_\theta$ | | | | |
| Horizon | | 4 | | |
| Activation | | ReLU | | |
| learning rate | | $5e^{-4}$ | | |
| number of hidden layers | | 2 | | |
| number of hidden units | | 256 | | |
| Polyak factor | | 0.995 | | |
| Learning step lagrangian multipliers | | $3e^{-4}$ | | |
| **Filter** | | | | |
| Filter Horizon | | 5 | | |
| number optimization steps | | 50 | | |
| learning rate | | $1e^{-1}$ | | |
| $\epsilon^{\text{safe}}$ | 0.5 | 0.5 | 0.5 | 2.5 |

Table 3: Hyperparameters for CASE

