# OpenReview forum: "Exploring Under Constraints with Model-Based Actor-Critic and Safety Filters"
_robot-learning.org/CoRL/2024/Conference — CoRL 2024_

### Official Review · Reviewer_1NEA · 2024-07-14
**A model-based reinforcement learning approach that uses safety filters to enhance safety at training time.**

**Originality:** 2
**Technical Quality:** 2
**Clarity Of Presentation:** 3
**Potential Impact:** 1
**Recommendation:** 2
**Confidence:** 4

**Review:**

This paper attempts to improve safety for model-based RL approaches, a topic that is relevant to robot learning. The paper is well-structured, with a clear indication of contributions and limitations. Unfortunately, the paper, in its current form, does not meet the standard of CoRL due to a significant lack of novelty, technical quality, and practicality to real-world robotics tasks. Detailed comments are attached below.

First, the mathematical definition of the unsafe state set (also the safe state set) $\mathcal{S}_{\text {unsafe }}= \\{s_t \mid \mathcal{I}\left(s_t, a_t\right)=1\\}$ is problematic. The action $a_t$ is used to define the indicator function. However, the author's definition implies that the classification of states as safe or unsafe is dependent solely on the state $s_t$​, without regard to the action $a_t$​.

Generally speaking, the proposed approach is a straightforward combination of existing works in MBRL (e.g., [12], [19]) and safety filters (e.g., Hsu et al., The safety filter: A unified view of safety-critical control in autonomous systems, which is not even cited). The few modifications to existing approaches are also minimal. For example, in (6) the authors propose to use safe samples when evaluating the objective, which is already a common practice in safe RL.

The title "Exploring Under Constraints" and subsection title "Constrained policy optimization" are misleading, since the safety objectives are incorporated using Lagrangian relaxation with a heuristic weight $\lambda$. Such formulation enjoys no guarantees as in constrained optimization and learning, and is ultimately unconstrained optimization, as in conventional RL.

The safety filter formulation encoded in (7) is very handwavy. The state-of-the-art safety filter approaches (e.g., HJ and CBF; again, see Hsu et al.) usually involve a backup policy that renders a safe set forward invariant. Why do the authors abandon those principled approaches for guaranteeing closed-loop safety, while resorting to some heuristic methods?

Regarding the experiments, the method is only evaluated with (simple) simulated benchmark examples - this is not acceptable for CoRL, especially given that the technical contributions of the paper are already on the weaker side. The author needs to provide either hardware demonstrations with real robots or a challenging example with simulations (e.g., humanoid locomotion, large-scale autonomous driving, or collaborative manipulation tasks).

Finally, the literature review is rather incomplete. The paper is closely related to safe RL with reach-avoid objectives, shielding (and its integration with safety-critical planning and/or learning), *learning* controlled forward invariance (e.g., Hamilton-Jacobi Reachability, control barrier functions), and safe learning of residual models. However, none of these is mentioned in the Related Work.

Minor issues:
* The full name of the acronym "CASE" was never mentioned.
* The fonts in Figure 3 and 4 are too small.

**Quality Of The Limitations Section:**

3

**Questions For Rebuttal:**

* What is the correct mathematical definition of safe/unsafe state sets? Why did the authors use indicator functions to define the safety objective? It has been shown to underperform the reach-avoid objectives that render dense rewards and much better safety performance (see, e.g., He et al., Agile but Safe, RSS 2024).
* What is the novelty of the paper aside from assembling existing approaches of MBRL and safety filters?
* What is a proper characterization of the proposed optimization formalism? Is it proper to call it "constrained optimization" while the actual problem being solved is unconstrained?
* Is the safety filter defined by (7) a principled approach to improve robustness in RL? Why did the authors choose not to use state-of-the-art safety filter methods such as HJ Reachability or CBFs?
* How does the proposed approach perform for realistic robotics tasks?

**Robotics Focus:**

2

**Summary Of Paper:**

This paper proposes a model-based reinforcement learning (MBRL) approach that leverages safety filters to improve safety at training time. This is in contrast to existing MBRL that does not explicitly account for safety during exploration. The proposed approach uses planning-based backup policies to lower constraint violations during training. The method is tested with simulated examples.

**Summary Of Recommendation:**

My recommendation is based on these weaknesses of the paper: lack of novelty, several technical issues, and dubious relevance to robotics. See my review for detailed justifications.

---

### Official Review · Reviewer_QmYe · 2024-07-21

**Originality:** 2
**Technical Quality:** 3
**Clarity Of Presentation:** 3
**Potential Impact:** 2
**Recommendation:** 3
**Confidence:** 4

**Review:**

1. The authors use a Gaussian network to approximate the dynamics model. However, it is unclear how the variances are utilized in the rollout. How is this uncertainty information leveraged during training?

2. The paper uses two $\lambda$ parameters to represent different quantities. It is advisable to use different notations to avoid confusion.

3. The authors only include the tail of the rollout in the optimization. How does this ensure that the midpoint along the rollout is safe? Is this choice simply to reduce the value of the multiplier, and would a smaller learning rate work just as well? Furthermore, is a threshold value $l$ missing in equation (6)? Since the constraint used here is an indicator function, the safety critic should be greater than 0. With the current formulation, i.e., $l=0$, the explosion of the Lagrangian multiplier is foreseeable.

4. From my understanding, the authors learn an ensemble of the value function, which is not parameterized as Gaussian. The model itself cannot distinguish between aleatoric and epistemic uncertainty. Instead, the safety filter uses the worst-case prediction instead of high epistemic uncertainty scenarios.

5. Since $V_{\text{max}}^{\text{safe}}(s_{t’})$ already captures the worst-case violations, why is a sum still needed in equation (8)?

6. Why does CASE have small violations at the very beginning? If no prior knowledge is given for the training, the dynamics and the value function will have poor estimations, and the policy should also contain certain violations until sufficient data is collected.

7. Omnisafe provides benchmarking results with 1e6 steps when the policy converges. However, the authors only show results with 1e5 steps, which is insufficient. The experiment shown in the paper is not convincing.

8. The experiments are evaluated exclusively on the Safe-Mujoco tasks, with no inclusion of robotic-related tasks.

[1] https://github.com/PKU-Alignment/omnisafe/tree/main/benchmarks

**Quality Of The Limitations Section:**

2

**Questions For Rebuttal:**

Please find them in the comments.

**Robotics Focus:**

2

**Summary Of Paper:**

This paper presents a model-based safe reinforcement learning (RL) method that incorporates a learned Markov Decision Process (MDP) and Model Predictive Control (MPC) to ensure safe exploration. The proposed method uses ensembles for each component to determine the worst-case scenario, which then triggers a safe MPC controller to ensure safety during exploration. The method is validated in the Safe-Mujoco tasks, showing fewer violations during the training process.

**Summary Of Recommendation:**

The paper presents a model-based safe RL approach. Integrating rollout, ensembles, and MPC for safe-critical problems are rational choices. However, the futher details of the experiments are needed to justify the claim.

---

### Official Review · Reviewer_Frme · 2024-07-21

**Originality:** 4
**Technical Quality:** 3
**Clarity Of Presentation:** 4
**Potential Impact:** 2
**Recommendation:** 3
**Confidence:** 4

**Review:**

Quality: Overall, the presentation of the paper is very clear and the paper is structured well.

Originality: The paper focuses on exploring using the model information alongside training the constrained policy which is a novel contribution. Their method also incorporates capturing epistemic uncertainty in the learned model.

Significance: The paper proposes a constrained model based reinforcement learning pipeline for reduced safety violations during exploration. However, they only compare the results with model-free baselines which I believe is not an apple-to-apple comparison and somehow doesn't highlight how their method is better than other model-based methods. In terms of runtime, the paper argues that average runtime of the proposed filter is around 50 Hz and therefore it can be tested on physical systems. However, the authors don't perform any hardware experiments. If the authors would have added hardware experiments, it could have shown proof of concept of their method in real world.

**Quality Of The Limitations Section:**

3

**Questions For Rebuttal:**

The paper claims that their proposed method helps in reducing constraint violations during training as compared to other constraint model-based optimization methods but in the results section they only compare their method with model-free baselines which is not what they claim in the abstract. How their methods compares with the Recovery RL paper for example should be shown. Also in "our contribution" section where discussion about Figure 1 is shown, there the authors claim comparison to CMBRL methods which is not the case.

The paper also uses only simple safety constraints such as constrained velocity rather than considering complex scenarios such as obstacle avoidance or multiple safety constraints. Please add few experiment with some additional safety constraints.

**Robotics Focus:**

3

**Summary Of Paper:**

The paper provides a novel approach for constrained model-based RL for reducing constraint violation during exploration by utilizing planning based safety filter as a backup policy. In addition, they utilize deep ensemble models to capture epistemic uncertainty in learned model thereby improving the ability to satisfy constraints. For instance, the filter is more often triggered in areas with high epistemic uncertainty leading to lesser safety violations during exploration.

**Summary Of Recommendation:**

Please read the review above.

---

### Author Rebuttal · Authors · 2024-08-11

We have revised our manuscript and incorporated the feedback from the reviewers. We are yet to add the results from our experiments on the safe-control-gym environments as the experiments are still ongoing. We will update our revised manuscripts once we have the results of these experiments.

---

### Decision · Program_Chairs · 2024-09-04

**Decision:**

Accept

**Comment:**

Reviewers agree that the problem solved is important and relevant for the robotics community. Unfortunately, the paper has three main weak points that should be handled in the rebuttal:
- The method seems to be not completely novel, and rather incremental. The authors should clarify the novelty of the approach.
- There seem to be some technical issues and unclear points in the proposed theory. Please refer to the reviewers' comments for more details. I expect a detailed answer to all their comments (and follow-up responses)
- There are no real robot experiments or interesting robotics tasks beyond simple Mujoco benchmarks. Most reviewers agree that this is a major drawback, and I'll not vote for accepting this paper if a more convincing benchmark is presented, to match the scope of the CoRL conference.

===

During the rebuttal, the authors managed to convince some of the reviewers about the technical soundness of their approach. Also, the authors added a reasonable robotics experiment (the 3d quadrotor stabilization).
While some reviewers argue that the novelty of the paper is a bit limited, it is overall an ok submission.
An important point to fix is to improve the description of the optimization (after equation 8)  and to add a safety critic baseline together with lagrangian approaches.
Also, I would require the reviewers to reference in the paper some of the works mentioned by reviewer 1NEA, specifically:

[1] Hsu, Kai-Chieh, Duy Phuong Nguyen, and Jaime Fernandez Fisac. "Isaacs: Iterative soft adversarial actor-critic for safety." Learning for Dynamics and Control Conference. PMLR, 2023.

[2] He, Tairan, et al. "Agile but safe: Learning collision-free high-speed legged locomotion." arXiv preprint arXiv:2401.17583 (2024).

[3] Hsu, Kai-Chieh, et al. "Safety and liveness guarantees through reach-avoid reinforcement learning." arXiv preprint arXiv:2112.12288 (2021).


I would mark this paper for acceptance, but I would not have any problem if the program chairs decide to reject this paper.